# New Labeling Rules for Wine: Wine Alcohol-Derived Calories and Polyphenol Consumption on Health

**DOI:** 10.3390/foods13020295

**Published:** 2024-01-17

**Authors:** Antoni Sánchez-Ortiz, Josep Maria Mateo-Sanz, Maria Assumpta Mateos-Fernández, Miriam Lampreave Figueras

**Affiliations:** 1Departament de Bioquímica i Biotecnologia, Facultat d’Enologia de Tarragona, Campus Sescelades, Universitat Rovira i Virgili, Marcel-lí Domingo, s/n, 43007 Tarragona, Spain; mariaassumpta.mateos@urv.cat (M.A.M.-F.); miriam.lampreave@urv.cat (M.L.F.); 2Departament d’Enginyeria Química, ETSEQ, Campus Sescelades, Universitat Rovira i Virgili, Marcel-lí Domingo, s/n, 43007 Tarragona, Spain; josepmaria.mateo@urv.cat

**Keywords:** bioavailability, polyphenols, alcohol intake, anthocyanins, procyanidins

## Abstract

Alcohol content, proanthocyanins and anthocyanins influence wine quality. The composition of wine depends on the type of cultivar, location, environmental conditions, and management practices. Phenolic compounds have attracted considerable research interest due to their antioxidant properties and potential beneficial effects on human health. However, the low bioavailability of anthocyanins creates a major bottleneck in their ability to exert beneficial effects. Despite extensive research on the effects of wine on human health, no clear evidence has been obtained on the benefits of wine quality or geographic area of production on health conditions, such as metabolic syndrome. Five climatically and geologically distinct wines were evaluated. Based on recent studies, meta-analyses, and pooled analyses of wine composition, along with the predicted low bioavailability of polyphenol compounds, we estimated the efficacy of five geographically distinct wines according to gastrointestinal absorption and the effects of alcohol intake on both men and women, with a view to ascertaining whether geographical origin influences the antioxidant serum composition of wine. Data on the estimated consumption of wine suggest that the polyphenol contents are similar regardless of choice of wine/area, while different alcohol compositions affect the level of alcohol and calorie intake. Thus, moderate wine drinkers should be advised to control the habit, but without exceeding the dose considered a healthy threshold (up to 30–40 g of alcohol/day in men and 10–20 g of alcohol/day in women), given no medical contraindications are present. These results will add value to the framework of the last reform of the Common Agricultural Policy (CAP) adopted in December 2021, where the European Parliament and the Council introduced new labeling rules for the wine sector and aromatized wine products.

## 1. Introduction

### 1.1. New Labeling Rules for Wine

The new Regulation (EU) 2021/2117 [1], published on 2 December 2021, amends the labeling rules for wines and aromatized wines, requiring the provision of all product data as defined in the food information regulation (EU) 1169/2011 [2]. From a practical standpoint, the primary objective of the new European wine labeling regulation is to ensure food safety for consumers and to secure their right to information on critical matters such as ingredient lists and nutritional tables. This regulation also aims to enhance transparency within the sector, fostering greater consumer trust. This new European regulation came into force on 8 December 2023.

Nevertheless, under specific conditions, it is permissible to provide consumers with this information through electronic labeling or e-label. If the operator wishes to indicate the ingredient list on an electronic label, they should nonetheless disclose the presence of allergenic substances on the physical label. If nutritional information is provided on the electronic label, the operator must indicate the energy value of 100 mL of wine or flavored wine on the physical bottle label (the option to use the energy symbol “E”). Alternatively, the operator may choose to display the ingredients and nutritional declaration on the packaging or on an attached label. The regulation provides for a transition period of 2 years. All wines produced and labeled after 8 December 2023 will be required to include the ingredient list and nutritional information.

In accordance with the consumer information regulation (EU) 1169/2011, the definition of an ingredient is any substance or product, including flavorings, food additives, and food enzymes, or any constituent of a compound ingredient used in the manufacture or preparation of a food product and remains present in the finished product, possibly in a modified form; residues are not considered ingredients. In this regard, Regulation (EU) 2019/934 [3], dated 12 March 2019, lists authorized enological practices, along with their classification, as additives or technological auxiliaries. Only additives are subject to labeling. The new labeling must indicate the category of the wine product and, if applicable, whether it is “dealcoholized” or “partly dealcoholized” wine. Furthermore, references to PDO or PGI, alcohol content, bottler or producer name, importer (if any), sugar content in sparkling wines, allergens, nutritional information, and the ingredient list should be included. All this information must be presented in a language understandable to consumers in the country where the wine is marketed, and the characters must be of a size equal to or greater than 1.2 mm.

Regarding nutritional information, the bottles must include both the energy value (in kJ and kcal) and the amounts of fats, saturated fats, carbohydrates, sugars, proteins, and salt. All of this should be expressed per 100 g or 100 mL and should appear within the same visual field of the label, using clear typography. While the energy value is compulsory on the physical label, the rest of the nutritional information may be provided via electronic means indicated on the packaging. Nutritional values are average values based on the manufacturer’s analysis, known average values, or generally established data. If the information is provided electronically, certain additional regulations must be met. User data cannot be collected or tracked, and a web address cannot be printed on the label. All wine ingredients must be listed in decreasing order of weight, and allergens, additives, and other specific elements have their own nomenclature and regulation.

This regulation is aimed at providing greater transparency to the wine industry and, above all, at better informing consumers. In a world increasingly concerned with healthy eating and transparency in the products they consume, this change was necessary and welcomed. The main challenge of this new regulation will be its implementation, as all wineries and wine producers will need to adapt their labeling and production processes to comply with the new regulations. However, while it represents a challenge, it is also an opportunity to stand out in the market and demonstrate commitment to consumer transparency and health.

The new European Regulation (EU) 2021/2117 will transform the wine industry and provide consumers with a better understanding of what they are consuming, which will be beneficial for all. The implementation of the new labeling regulations carries important implications for consumers’ understanding of the precise meaning of each component and ingredient, as well as the associated nutritional information. In the interest of providing more detailed disclosure, this article aims to enhance consumers’ understanding of nutritional analyses, providing them with a more comprehensive perspective of the elements present in the final product. By detailing the presence and function of additives more precisely and by presenting relevant nutritional information, the goal is to empower consumers to make informed decisions about their wine consumption, aligning it with their personal preferences and dietary needs. Furthermore, in this study, wines from different geographical areas were analyzed to assess whether geographical origin significantly influences the composition of wine and the calorie intake to better understand the impact of the nutritional facts of the new labeling.

### 1.2. Spanish Regulatory Wine Classification, Winemaking Procedure, and Common Additives in Wine

In European wine law, wines are classified into several categories, each indicating different levels of quality and geographical specificity. Table wines, known as “Vino de Mesa” in Spain, are the most basic category, characterized by their simplicity and lack of specific requirements. Land wine, referred to as “Vino de la Tierra” in Spain, denotes a specific geographical region and may offer slightly higher quality than table wines. Protected Geographical Indication (PGI) wines, such as France’s “Vin de Pays” or Italy’s IGT wines, are associated with specific regions and must adhere to strict production criteria, offering consumers an assurance of quality and provenance. Finally, wines labeled as Denominación de Origen (DO), Denominazione di Origine Controllata (DOC), Denominazione di Origine Controllata e Garantita (DOCG) in Italy, or Appellation d’Origine Contrôlée (AOC) in France are considered of the highest quality and are rigorously regulated in terms of grape variety, production methods, and geographical origin, providing consumers with a guarantee of excellence and adherence to strict standards. Each category represents an incrementally higher level of quality and geographic specificity, offering valuable information about the wine’s origins and characteristics.

The winemaking process comprises several critical steps. (i) Harvesting: the grapes are carefully harvested at the optimal time, ensuring the desired level of ripeness and sugar content. (ii) Crushing and pressing: The harvested grapes are crushed to extract their juice. Typically, white wines are promptly pressed, whereas red wines may undergo maceration with the grape skins to extract color and tannins. (iii) Fermentation: Yeast is introduced to the grape juice to initiate the fermentation process, converting sugars into alcohol. This transformation can occur in stainless steel tanks, oak barrels, or other containers. (iv) Aging: Following fermentation, the wine may undergo aging in oak barrels or stainless-steel tanks to enhance its flavor and character. The duration of this aging process varies according to the desired wine style. (v) Clarification and stabilization: to achieve microbiological stability and remove remaining solids, the wine undergoes processes such as fining, filtering, and cold stabilization. (vi) Bottling: Once the wine has been clarified and stabilized, it is bottled and sealed. At this stage, winemakers may opt to incorporate specific additives, such as sulfur dioxide, to aid in wine preservation and prevent spoilage.

It is important to note that the use of additives in winemaking is carefully regulated, and winemakers must adhere to specific guidelines regarding which additives are permitted and in what quantities. These additives are used judiciously to maintain the quality and stability of the wine. Common additives in winemaking include the following: (i) sulfur dioxide: This is used as a preservative to inhibit the growth of undesirable microorganisms and to prevent oxidation. The maximum allowable levels of sulfur dioxide in wine are regulated by various government authorities, such as the US FDA in the United States and the EU regulations in Europe. (ii) Yeast nutrients: These are used to support healthy fermentation by providing essential nutrients for yeast metabolism. The specific types and maximum doses of yeast nutrients may vary depending on the winemaking practices and regulations in different regions. (iii) Fining agents: These are used to clarify the wine by removing particulate matter. Examples of fining agents include bentonite, egg whites, and various types of proteins. The types and maximum allowable doses of fining agents are typically regulated by wine authorities to ensure the safety and quality of the final product. It is important to note that regulations regarding additives in wine can vary by country and region, and winemakers must comply with specific guidelines set forth by their local regulatory bodies. Additionally, winemakers often follow industry best practices and standards to ensure that the use of additives is carefully monitored and remains within acceptable limits. For specific information on the maximum allowable doses of additives in wine, the authors recommend referring to the official regulatory agencies governing winemaking practices in your specific region. Additionally, consulting with a qualified enologist or regulatory expert can provide detailed insights into the specific regulations and guidelines applicable to winemaking and the use of additives.

Considerations of caloric intake become particularly important when consuming wine, especially for individuals managing conditions such as metabolic syndrome. This is attributed to the potential impact on overall caloric intake and metabolic health. Notably, wine, like many other alcoholic beverages, contains calories primarily from alcohol, with sweet wines also contributing calories from residual sugars. In the context of metabolic syndrome, which encompasses a cluster of conditions including high blood sugar, excess abdominal fat, abnormal cholesterol or triglyceride levels, and high blood pressure, vigilance in monitoring caloric intake is paramount. Excessive calorie consumption, particularly from alcohol, can contribute to weight gain and exacerbate metabolic syndrome. From a scientific perspective, the caloric content of wine, approximately 7 calories per gram of alcohol, plays a significant role in overall energy intake. Excessive consumption can disturb the balance between energy expenditure and intake, thus potentially contributing to weight gain and metabolic dysregulation.

Following the implementation of the new wine labeling regulations, which include the disclosure of the caloric content in a serving of wine, consumers will be empowered to comprehend the significance of understanding the caloric load of a standard glass of wine in the context of their total daily caloric intake. These data hold relevance for individuals managing conditions such as metabolic syndrome, as it equips them to make informed dietary decisions, including those related to wine consumption, in consideration of their specific health conditions. For individuals with metabolic syndrome, meticulous attention to the caloric content of wine and other dietary sources is vital to uphold a meticulously balanced and health-conscious diet. Modulation of alcohol consumption and overall caloric intake, in conjunction with an emphasis on nutrient-dense dietary options, is frequently advised for individuals managing metabolic syndrome. It is important to note that the influence of wine consumption on metabolic health is influenced by individual factors such as comprehensive dietary patterns, levels of physical activity, and any co-existing medical conditions. Seeking guidance from a healthcare professional or a qualified dietitian can facilitate tailored recommendations for navigating dietary choices, encompassing the consumption of wine, in the context of managing metabolic syndrome.

### 1.3. Grape Polyphenol Synthesis, Structure, and Composition

Two main pathways are implicated in the biosynthesis of phenolic compounds: shikimic acid and malonic acid [4]. The malonic acid pathway is considered one of the most important sources of phenols in fungi and bacteria but is less extensively used in superior plants. On the other hand, the shikimic acid pathway is responsible for the biosynthesis of most polyphenolic compounds in plants. Starting from erythrose-4-phosphate and phosphoenolpyruvic acid, a sequence of reactions is initiated, leading to the generation of shikimic acid and several aromatic amino acids (phenylalanine, tryptophan, and tyrosine). Most polyphenolic compounds are derived from phenylalanine. Phenolic compounds are important contributors to the antioxidant properties and the color and mouthfeel of red wine [5]. Two important families of polyphenol compounds present in grapes are known to influence final wine quality, specifically, proanthocyanins (condensed tannins) (Figure 1) and anthocyanins (Figure 2). The former contribute to the astringency and bitterness of wines, while the latter are pigments responsible for wine color [6,7]. Polyphenol composition is attributed not only to the type of cultivar but also to the location of grapes, environmental and management practices, and the growing season [8,9,10]. Proanthocyanins and anthocyanins constitute the two most abundant classes of phenolic compounds in berry skin. Condensed tannins are polymeric flavan-3-ols, mainly comprising subunits of (-)-epicatechin, in addition to significant amounts of epigallocatechin, (+)-catechin, and epicatechin-3-O-gallate [11].

Anthocyanins are responsible for the color of red and black varieties of grapes. Most Vitis Vinifera varieties produce non-acylated glucoside, acetyl glucoside, coumaroyl glucoside (and, to a lesser extent, caffeoyl glucoside) derivatives of delphinidin, cyanidin, petunidin, peonidin, and malvidin. Each variety of grape has a specific anthocyanin profile. Anthocyanin analysis has been proposed for varietal authentication of grapes and wines. Both anthocyanins and tannins are partially extracted from grape skin during winemaking and undergo structural transformations through several reactions with significant influence on wine sensory characteristics due to their involvement in astringency, bitterness, color intensity, and color stability [11,12].

Anthocyanins represent the largest group of water-soluble pigments in the plant kingdom. These compounds are widely distributed in crops, beans, fruits, vegetables, and red wine, resulting in the human ingestion of significant amounts of anthocyanins from plant-based daily diets. In general, anthocyanin pigments are stable under acidic conditions but are unstable and rapidly broken down under neutral conditions.

Focusing on phenolic acids (Figure 3), they are predominantly colorless molecules present in red wines. Their presence does not contribute specific aroma or flavor to the wine; however, they play an important role in the chromatic evolution of wine, potentially contributing to the yellowish tone of older red wines if oxidation occurs.

Another large group of flavonoids are flavonols (Figure 4) (quercetin, myricetin, kaempferol, isorhamnetin and their glycosides), which contribute to bitterness, red wine color [13], and antioxidant activity [14]. The concentration of phenolic compounds in grapes is also dependent on the grape cultivar and influenced by viticultural and environmental factors, such as maturity stage, seasonal conditions, production area and fruit yield [15,16,17,18].

Resveratrol is synthesized in grape skin as a response to fungal infection (Figure 5). The compound acts as a phytoalexin, preventing pathogen proliferation. During an attack of Botrytis cinerea (the main fungal infection damaging wine crops), plants form a resveratrol barrier [19]. Additionally, in grape berries of some varieties, piceid, a stilbene glucoside of resveratrol is detected, which is related to the biosynthesis of resveratrol. Together with resveratrol, its oligomers (the dimer trans-ε-viniferin and trimer α-viniferin) have been detected in wine (Figure 6). Resveratrol levels in red wines range between 0.1 and 14.3 mg/L [20].

Table 1 and Table 2 present the main grape and wine phenolic antioxidants (including phenolic acids) and their classification [15,18,21,22,23].

### 1.4. Effects of Climate, Soil, and Vineyard Location on Grape Composition

Climate and wine quality are strongly linked in viticultural areas worldwide. Given that the climate is sufficiently warm to ripen a specific grape cultivar, quality is inversely related to warmth and the length of the summer [5]. A few studies on the climatic effect on quality suggest that wines derived from cooler climates are fresher, more acidic, and finer in bouquet and aroma, while wines from warmer regions are higher in alcohol content and lacking in taste and aroma. Vines absorb only water and dissolved mineral ions from the soil, and a poor soil structure will allow grapevines to send roots downward. Priorat soil types are typically found on slopes and ridges with characteristic erosion and decomposition. Vines are planted on slate-driven soils and are often dry-farmed, receiving little or no irrigation. On the other hand, valley soils are typically more fertile and denser, composed of finer textural elements [24]. Deep penetration of root systems into these soils can lead to excessive growth at the expense of concentrated flavors. Priorat AOC, which is situated behind the coastal mountain range of Tarragona, is characterized by a Mediterranean climate [25] with very little precipitation during the vegetation cycle. The soil is of poor quality, dry, and pebbly, primarily composed of slate.

### 1.5. Health Effects of Wine Polyphenols

The phenolic components of wine have garnered significant research interest due to their antioxidant properties and potential beneficial effects on human health [26]. Grape seed extract has been commonly utilized in recent years as a nutritional supplement [27]. However, the analysis of phenolic compounds from vine and wine products (grape seeds and skins, musts, and wines) is complex due to their significant diversity. The dietary intake of polyphenols from red fruits, vegetables, and red wine can reach up to 200 mg/day, and their consumption via red wine has been proposed as part of the reason underlying the “French Paradox” [28]. This suggests that a diet rich in saturated fats and moderate alcohol consumption could prevent the elevated levels of heart disease, cancer, and stroke found in other countries. Anthocyanins are effective antioxidants [29] and possess other biological activities with health benefits independent of antioxidant capacity. This includes inhibition of cancer cell growth in vitro [30], induction of insulin production in isolated pancreatic cells [31], a reduction in starch digestion by inhibiting α-glucosidase activity [32], suppression of inflammatory responses [33], and protection against age-related decline in cognitive behavior and neuronal dysfunction in the central nervous system [34]. The breeding of crops with increased anthocyanin content has been a significant target of research [35]. However, to achieve biological effects in specific tissues or organs, anthocyanins must be bioavailable, meaning they are effectively absorbed from the gastrointestinal tract (GIT) into the circulation and delivered to the appropriate locations within the body. Studies on the oral administration of anthocyanins have confirmed the increased antioxidant status of serum [36,37], but this is usually accompanied by a very low uptake of anthocyanins and corresponding low levels of urinary excretion as intact or conjugated forms. The apparent low bioavailability of anthocyanins casts doubts on the ability to exert their proposed beneficial effects in the human body. Therefore, anthocyanins are not generally recognized as a physiological functional food factor. However, cyanidin 3-glucoside (C3G), a typical anthocyanin, is reported to exert antioxidative and anti-inflammatory effects in vitro and in vivo [38,39,40,41,42], clearly suggesting beneficial effects beyond its antioxidant capacity. Epidemiologic studies have linked flavonoid-rich foods with a reduced risk of cancer and cardiovascular disease. While the mechanisms underlying the suggested health benefits of flavonoid-rich foods remain to be fully elucidated, in vitro and in vivo studies have demonstrated that flavanols and procyanidins from wine have several beneficial biological activities, including the ability to reduce oxidative damage, promote endothelium-dependent relaxation, and decrease platelet aggregation.

#### 1.5.1. Metabolic Syndrome

Metabolic syndrome is a combination of several clinical features including central obesity, high blood pressure, and elevated fasting glucose and triacylglycerol contents, along with low concentrations of HDL cholesterol, and insulin resistance. The clustering of these features is speculated to increase the risk of cardiovascular disease, which is associated with each component. Consistent with this theory, recent studies have reported that metabolic syndrome markedly increases cardiovascular morbidity and mortality. Metabolic syndrome components include the following: (1) central obesity measured as waist circumference (102 cm for men and 88 cm for women), (2) high serum triacylglycerol (150 mg/dL), (3) low serum HDL cholesterol (40 mg/dL for men and 50 mg/dL for women), (4) hypertension (systolic/diastolic pressure of 130/85 mmHg, and (5) high fasting glucose (110 mg/dL). Metabolic syndrome is defined as the presence of three or more of these components.

#### 1.5.2. Alcohol Intake

Limited studies to date have focused on the effects of alcohol on the development of metabolic syndrome. While an association between alcohol drinking and prevalent metabolic syndrome has been documented, the findings are inconsistent. Some studies indicate that the relationship is inversely linear, J-shaped, or positively linear, whereas others show no association. In addition, the association appears to differ based on type of alcoholic beverage. Compared with no alcohol consumption [43], light to moderate drinking of wine and beer appears favorable for reducing the prevalence odds ratio of metabolic syndrome, whereas liquor drinking tends to increase the ratio or have no association with metabolic syndrome. Earlier studies on the association between alcohol consumption and metabolic syndrome have had limited success in establishing causality owing to their cross-sectional design. To evaluate the effect of alcohol on the development of metabolic syndrome, the incidence of metabolic syndrome was prospectively examined in relation to alcohol consumption status, including average daily amount consumed, type of alcoholic beverage most consumed, and drinking frequency [44]. Additionally, a prospective study on a Korean cohort aged 40–69 years showed that heavy drinking, particularly liquor, is associated with an increased risk of metabolic syndrome by affecting its components, including waist circumference, triacylglycerol content, blood pressure, and glucose. Although mounting evidence strongly supports beneficial cardiovascular effects of moderate red wine consumption (one to two drinks per day; 10–30 g alcohol) in most populations, clinical advice to abstainers to initiate daily alcohol consumption has not yet been substantiated in the literature and must be considered with caution on an individual basis [45]. Further research is warranted to clarify the association between the level of alcohol consumption and metabolic syndrome risk, as well as the beverage-specific association in terms of beer or wine.

#### 1.5.3. Coronary Heart Disease (CHD)

Coronary heart disease (CHD), also known as coronary artery disease, is the narrowing of the small blood vessels that supply blood and oxygen to the heart. CHD is usually caused by atherosclerosis, which occurs when plaque builds up on the walls of arteries, leading to narrowing. With the narrowing of coronary arteries, blood flow to the heart can slow down or stop, causing chest pain (stable angina), shortness of breath, heart attack, and other symptoms. Cardiovascular disease is the main cause of mortality in industrialized countries, but incidence rates show marked geographical differences. The low incidence of CHD in Mediterranean countries has been partly ascribed to dietary habits. Recent findings from studies on a large European cohort suggest that a high degree of adherence to the Mediterranean diet is associated with a reduction in mortality. In small-scale clinical studies, the Mediterranean diet or some of its components have been linked to reduced blood pressure along with improved lipid profiles [46] and endothelial function. High blood pressure (HBP) is a serious condition that can trigger CHD and other health problems. Blood pressure refers to the force of blood pushing against the walls of arteries as the heart pumps out blood. A consistent increase in blood pressure over time can damage the body in many ways. Alcohol intake from any type of alcoholic beverage appears beneficial, but some studies suggest that red wine confers additional health benefits. The benefits of red wine are further supported by a meta-analysis of 13 studies involving 209,418 participants that showed a 32% risk reduction in atherosclerotic disease with red wine intake, which was greater than 22% risk reduction upon beer consumption. The dietary intake of flavanones, anthocyanidins and specific foods rich in flavanoids is potentially associated with a reduced risk of death due to cardiovascular heart disease.

Conversely, other investigations have failed to demonstrate the beneficial effects of red wine, leading to the conclusion that additional lifestyle factors, such as diet, exercise, socioeconomic status, or pattern of alcohol consumption potentially play a role in the lower rates of atherosclerosis in wine drinkers [47,48]. However, increased alcohol consumption for the purpose of cardioprotection cannot be justified. There is no rational reason for non-drinkers to start consuming wine as a preventive measure considering that several other well-proven therapies exist for cardiovascular risk reduction, such as exercise, smoking cessation, blood pressure control, and the lowering of cholesterol [49].

#### 1.5.4. Dyslipidemia and Diabetes

High-density lipoprotein (HDL) is one of the five major groups of lipoproteins (chylomicrons, VLDL, IDL, LDL and HDL) that facilitate transport of lipids, such as cholesterol and triglycerides, within the water-based bloodstream. In healthy individuals, ~30% blood cholesterol is carried by HDL. HDL is proposed to remove cholesterol from atheroma within arteries for transport to the liver for excretion or re-utilization. Therefore, HDL-bound cholesterol is sometimes known as “good cholesterol” or HDL-C. A high level of HDL-C could protect against cardiovascular diseases and, conversely, low HDL cholesterol levels (<40 mg/dL or ~1 mmol/L) increase the risk of heart disease. Cholesterol contained in HDL particles is considered beneficial for cardiovascular health, in contrast to “bad” LDL cholesterol.

Recent attention has focused on food factors that may be beneficial in preventing body fat accumulation and reducing the risk of diabetes and heart disease. Although several drugs that target obesity-related metabolic diseases or prevent body fat accumulation have been marketed, little evidence showing that food factors are directly beneficial in improving the dysfunction of adipocytes responsible for adipocytokine secretion and lipid metabolism is available [50]. Anthocyanins were recently shown to enhance adipocytokine (adiponectin and leptin) secretion, and the expression of PPARγ and adipocyte-specific genes in isolated rat adipocytes without stimulation via PPARγ ligand activity for the first time [51]. However, other anthocyanin-responsive genes may exist that would contribute to clarification of the biological basis for utilization of anthocyanins as physiological functional food factors. Nutrigenomics is the application of high-throughput genomic tools in nutrition research. Significant advances in DNA microarray technology should promote our understanding of anthocyanin-mediated influence on gene expression and regulatory mechanisms of genes responsible for the prevention of obesity and amelioration of insulin sensitivity through modulation of adipocyte function. Data from DNA microarray analysis showed for the first time that anthocyanins enhance the lipolytic activity and gene expression of related enzymes in adipocytes [52]. Dietary anthocyanin has recently been shown to significantly suppress the development of obesity. A few studies suggest that anthocyanins regulate obesity and insulin sensitivity associated with adiponectin and leptin secretion and PPARγ activation in adipocytes.

The normal non-diabetic blood glucose level ranges from 70 to 110 mg/dl depending on the type of blood tested. Glucose level > 140 mg/dL is usually indicative of diabetes (except in newborns and some pregnant women). Insulin, a hormone made by the pancreas, helps the body utilize glucose for energy. Insulin resistance is a condition in which the body produces insulin but cannot use it properly. In individuals with insulin resistance, the muscle, fat, and liver cells do not respond normally and require more insulin for glucose entry into cells. Eventually, the pancreas fails to keep up with the body’s surplus need for insulin. Excess glucose builds up in the bloodstream, setting the stage for diabetes. Patients with insulin resistance often have high levels of both glucose and insulin circulating in the blood. 

Insulin resistance [53] increases the risk of developing type 2 diabetes and heart disease. Atherosclerotic diseases are prevalent as secondary complications associated with type 2 diabetes, and a diet high in readily absorbable carbohydrates is associated with increased risk of type 2 diabetes [54]. Accumulating epidemiologic data implicate postprandial hyperglycemia as a risk factor in the development of cardiovascular disease. Elevated postprandial glucose levels may have a direct toxic effect on the vascular endothelium mediated via oxidative stress, independent of other cardiovascular risk factors, such as hyperlipidemia [55]. Postprandial hyperglycemia may also exert effects through its substantial contribution to total glycemic exposure. Ischemia-reperfusion causes oxidative damage that is enhanced with repetitive postprandial hyperglycemia [56]. Among the cells damaged by diabetes are primary sensory neurons, also known as dorsal root ganglion neurons. Damage to these cells triggers diabetic peripheral neuropathy. Elevated glucose leads to apoptosis in neurons [57] accompanied by increased oxidative stress. Procyanidins have insulin-like effects in insulin-sensitive cells that could explain their antihyperglycemic effect in vivo. These effects, in addition to their antioxidant activity, may contribute to beneficial effects against diabetes [58]. Earlier epidemiologic studies indicate that alcohol consumption is associated with improved insulin sensitivity but experimental evidence to confirm this finding is limited. For instance, moderate wine consumption by overweight women in a previous study did not improve or impair insulin sensitivity or induce changes in any of the known indicators of insulin sensitivity, including body weight and composition, blood lipid, and blood pressure [59].

### 1.6. Bioavailability of Anthocyanins and Procyanidins

Bioavailability refers to the amount of a specific nutrient in food or a bioactive ingredient ultimately used by the body to perform specific physiological functions, becoming available at the site of action after absorption from the gastrointestinal tract [58]. Several factors influence nutrient bioavailability, including digestion, absorption, distribution in blood and entry into tissue where it is physiologically effective. Bioavailability can be quantified to some extent by measuring (1) the amount of the nutrient in various body tissues and fluids or (2) the growth or enzyme activity dependent on the nutrient. However, a nutrient is rarely stored in a single body tissue, and therefore, determining the levels in single tissues does not accurately reflect true bioavailability [60]. For example, the levels of nutrients in blood, which are easily accessible for measurement purposes, may not reflect those in other tissues that serve as major stores, such as the liver. Each step involved in the process that facilitates bioavailability of nutrients is affected by a variety of factors in food and the nutritional status of individuals. It is particularly difficult to assess bioavailability in cases where the nutrients are present in many different forms in foods and tissues. 

While the flavanol monomers in wine (-)-epicatechin and (+)-catechin) are readily absorbed and metabolized in humans [61], little is known about the bioavailability and metabolism of procyanidins. Several studies have shown rapid absorption of polyphenolic compounds, such as procyanidins, quercetin and flavanols, from grapes into plasma [62,63,64]. After two weeks of daily red wine consumption (375 mL), total plasma phenol concentrations increased significantly and trace levels of metabolites from (+)-catechin and (-)-epicatechin were detected in plasma. However, the biotransformation and bioavailability patterns of many dietary polyphenols remain to be clarified, particularly those of anthocyanins [65] The tissue distribution and biotransformation pathways for several dietary polyphenols are yet to be determined. Furthermore, the biological activities of metabolites of many dietary polyphenols require further investigation. The potential health benefits of dietary polyphenols require confirmation in both animal models of disease and humans at appropriate doses. While in vitro studies have provided insights into the mechanisms of action of individual dietary polyphenols [66], the significance of these findings requires validation in vivo. Further efforts should be made to integrate the available in vitro and in vivo activity data with bioavailability data for assessing the potential utility of various dietary polyphenols. Accumulating evidence from human feeding studies suggests that the absorption and bioavailability of specific flavonoids are markedly higher than originally believed [67]. Most flavonoids in plants are attached to sugars (glycosides) and occasionally exist as aglycones. Aglycones are freely absorbed from the gut through passive diffusion, while glycosides are hydrolyzed by colon microflora before gastrointestinal absorption [67].

Human feeding trials with wine have demonstrated that procyanidins can survive the acidic milieu of the stomach and are therefore not initially broken down, entering the small intestine intact. Consistent with this finding, dimer B2 [epicatechin (4β-8)-epicatechin] was detected in human plasma as early as 30 min after consumption of a flavanol-rich food. Thus, while the metabolic fate of dimer B2 is yet to be elucidated, clearly it can be absorbed, supporting a contributory role to the benefits of flavanol/procyanidin-rich food [68]. In terms of absorption, lactase phlorizin hydrolase (LPH) hydrolyzes the majority of anthocyanidins, allowing for absorption by the small intestine [69]. Notably, cyanidine-3-glucoside is not hydrolyzed in the small intestine [70]. Other recent studies indicate that bilitranslocase plays a role in absorption at the gastric level [71]. The degradation of anthocyanins mainly takes place in the intestine, whereas both the intestinal microbiota and pH play important roles in catabolizing anthocyanins into metabolites. The degradation products of anthocyanins in the gastrointestinal tract are reported to be phenolic acids, phenol aldehydes and phenols. Both anthocyanins and degraded products or metabolites can be absorbed through either passive diffusion or active transport. The molecular absorption mechanism is still to be fully clarified to assess the real in vivo digestion, absorption, bioavailability, and bioactivities of anthocyanins [72].

### 1.7. Synergy of Wine Polyphenols with Food

The assessment of nutrient bioavailability remains critical to our understanding of the mechanisms through which humans utilize essential nutrients from consumed foods and how foods satisfy nutritional requirements [73]. Different food components could reduce or enhance nutrient bioavailability. Some components form complexes with a nutrient and prevent its digestion or absorption or even induce degradation. In the case of wine, phenolic compounds can chelate iron, and red wine decreases the concentration of digested phenolic products attributable to the formation of iron-polyphenol chelates. In terms of protein affinity, flavonoids are strongly affected by the presence of milk, especially after the digestion process [74]. Additionally, using an in vitro digestion procedure, [75] found that co-digestion of red wine with vitamin C and meat resulted in an increase and decrease in antioxidant capacity and total phenol content, respectively. Similarly, co-digestion of raspberry extract with meat, bread and cereals decreased the recovery of total phenol [76] but not of anthocyanins in the serum-available fraction. The hydroalcoholic matrix of wine could facilitate the solubility and absorption of its phenolic components [77].

## 2. Materials and Methods

### 2.1. Sampling and Winemaking

The climate in the Priorat region (Tarragona, Spain), characterized by very high temperatures during summer, drought, and steep and poor stony soils, promotes ecosystem vulnerability to current global changes. A recent report on how mesoclimate influences wine quality has shown the differences between Priorat mesoclimates [5]. The study involved five different vineyards, whereby two villages under two different mesoclimates were selected (early and late ripening and two/three different parcels in each mesoclimate, topographically located up or down the slope). At each of the two municipalities, El Molar (early) and Porrera (late), 60-year-old vines were selected, planted in bush at a density of 5000–6000 vines·ha^−1^. To evaluate the effects of topography and mesoclimate on the qualitative potential of *Vitis vinifera* cv Carignan in the Priorat region, [5] monitored the evolution of the maturity process in the five parcels and analyzed the composition of both the grapes grown and wines obtained. 

The grapes were fermented after three days of cool maceration for color extraction, followed by fermentation of all reducing sugars, addition of 20 g/hL sulfur dioxide to preserve oxidation, and finally bottling. The wine did not undergo malolactic fermentation and was therefore young, without oaking or aging. OIV methods (International Organization of Vine and Wine) were used to analyze alcohol by volume (ABV), total tartaric acidity (ATT), pH, total anthocyanins [21], DMACA (flavan-3-ol by derivatization with p-dimethylaminocinnamaldehyde) [23], and total tannins in wine. ANOVA was performed using the general linear model procedure. The Tukey test was used for post hoc analysis (XLSTAT statistical package, EXCEL) between plots.

Alcohol by volume (ABV), pH, TA, T Ant, and tannins were analyzed. Anthocyanin identification followed the methodology detailed in [22] through high-performance liquid chromatography (HPLC) using a Hewlett Packard Liquid Chromatograph (Waters Corporation, Mildford, MA, USA) equipped with a Zorbax Eclipse Plus C18 Column (150 × 2.1 mm; 3.5 µm) and a Zorbax Eclipse Plus-C18 Precolumn (12.5 × 4.6 mm; 5 µm). The injection volume was 5 µL; elution was performed with a mobile phase A of HPLC-grade water (0.2% trifluoroacetic acid) and a mobile phase B using methanol (0.2% trifluoroacetic acid). The column temperature was set at 50 °C and the HPLC was coupled to a Diode Array Detector (DAD). A mass spectrometry (MS) detector was used to assist in the identification. Anthocyanidin-3-monoglucosides and respective acetylated and coumaroylated glycosides were identified based on their ultraviolet–visible (UV–vis) spectra and retention times [5].

Anthocyanidins were quantified, making a comparison with internal standards. Calibration curves were obtained by injecting standards with different concentrations of malvidin 3-glucoside (Extrasynthese, Genay, France). The range of the linear calibration curves was 0.1 to 1.0 mg/L for the lower (R^2^ > 0.996), 0.1–5.0 mg/L for intermediate (R^2^ > 0.987), and 10.0–200.0 mg/L for the higher concentration compounds (R^2^ > 0.987). Unknown concentrations were determined from the regression equations, and the results were expressed as milligrams of malvidin 3-glucoside. Free anthocyanin content was determined using a calibration curve (based on peak area, y = 0.7968x + 7.5756; R^2^ = 0.9774), which was established using malvidin 3-glucoside standard solutions submitted to the same procedure. The repeatability of HPLC analysis gave a coefficient of variation of <7%.

Procyanidins were analyzed by injecting 3 µL of wine samples through Rapid Resolution Liquid Chromatography (RRLC) using a Zorbax Eclipse XDB-C18 (50 × 30; 1.8 µm) followed by an RRLC in-line pre-column (4.6 mm, 0.2 µm) at 30 °C. The HPLC injection volume was 1.4 µL, with a 0.7 mL/min flux; mobile phase A: water (0.1% formic acid), mobile phase B: methanol (0.1% formic acid). Phenolic compounds were identified according to their order of elution, retention times of pure compounds (gallic acid, catechin, procyanidin dimer B2, mono gallate dimer, procyanidin trimer C1, and epicatechin gallate) and their molecular masses. A different calibration curve was used for gallic acid (R^2^ = 0.9957), catechin (R^2^ = 0.9779), Procyanidin B2 (R^2^ = 0.9851), Epicatechin (R^2^ = 0.9884), Epicatechin gallate (R^2^ = 0.9935) and Trimer C1 (R^2^ = 0.9848).

### 2.2. Composition of Priorat Wines

To assess the estimated consumption of 5 wines, a previous analysis of individual wines was conducted. These wines were not blended with other grapes nor with vintages; thus, the 5 wines are considered single varietals, vineyard, and vintage. This adds a lot of value to this review as most of the wines on the market are difficult to prove as being made from a single blend (European law allows labeling as single grape and single varietal even if this contains a minimum of 15% of other grapes and other vintages in the same blend). In our review, total anthocyanin, and tannin contents, DMACH, pH, total acidity, and alcohol % of each wine sample are presented in Table 3. 

### 2.3. Statistical Analysis

Statistical analysis was conducted with SPSS, version 17.0 (SPSS Inc. 233 Chicago, IL, USA). ANOVA was performed for analysis of wine composition and HPLC-mediated determination of procyanidins and anthocyanins. The Tukey test was applied for post hoc analysis. Data processing was developed in two steps: independent evaluation of the five sites (1, 2, 3, 4 and 5) and evaluation of the results based on the effect of geographical origin. In this case, two groups of wines were generated: (E) early ripening geographical area = Sites 1 and 2; (L) late ripening geographical area = Sites 3, 4 and 5. E corresponds to an average of both wines from the early region (El Molar) and L to an average of the three wines from the late region (Porrera).

## 3. Results

### 3.1. Anthocyanin and Procyanidin Analysis of the Wines

According to the analysis of five wines from five different geographical areas in the Priorat described by [5], wines from Site 1 and Site 2 regions represent wines derived from the warmest areas (early ripening, E), while wines from Site 3, Site 4, and Site 5 are obtained from the coldest areas (late ripening, L). The wines are designated as Site 1 (El Molar, early region, uphill), Site 2 (El Molar, early region, downhill), Site 3 (Porrera, late region, downhill west-exposed), Site 4 (Porrera, late region, downhill east-exposed), and Site 5 (Porrera, late region, uphill). Total anthocyanin contents of two wines (Site 1 and Site 4) were not significantly different, while the wines from the Site 2, Site 3, and Site 5 locations displayed higher anthocyanin concentrations compared to those of Site 1, which had the lowest content. Wines from Site 1 had the highest concentration of tannin, which was significantly different from the tannin contents of wines from Site 2, Site 3, Site 4, and Site 5. Thus, the wine produced from Site 1 contained the highest tannin and lowest anthocyanin amounts. The pH for Site 1 appeared lower than that for Site 2 and Site 3, but not to a significantly different extent. In contrast, the pH differences relative to Site 4 and Site 5 were marked. The highest differences in total acidity were observed between Site 1 and Site 5 wines that belonged to different mesoclimatic areas. The alcohol content was markedly higher in Site 1 compared to Site 3 and Site 4 (an increase in 1.56% alc. vol in Site 1 vs. Site 4 and 1.26% alc. vol in Site 1 vs. Site 3). Owing to several differences among the five wines selected for the study, a new statistical group was evaluated. Wines from the early grape-ripening El Molar area were considered (E = Site 1 + Site 2) and those from the late grape-ripening Porrera region (L = Site 3 + Site 4 + Site 5) grouped together. As a result, two different regions and only two treatments or wines were also considered. The results from the chemical analyses were significantly different between the two early ripening geographical areas (Site 1 and 2) with a *p*-value < 0.05. Considering the influence of only early and late regions instead of vineyard location, no marked differences in total anthocyanins, pH, and total acidity were observed. In contrast, total tannin content, DMACH, and alcoholic degree (% alc. by vol.) were significantly different, while no major differences were recorded in total anthocyanin contents between the treatments in the first analysis of wine (Table 2). The results from the HPLC analysis were used for the evaluation of specific anthocyanins and procyanidins, with a view to determining the polyphenol types that are more abundant in different wines with potential health benefits. Table 4 and Table 5 show the previous results obtained from the five different wine-growing areas in Priorat.

HPLC analysis of the five wines revealed 15 anthocyanins, mainly 3-0-glucosides of malvidin, petunidin, delphinidin, peonidin and cyanidin (Table 4). Acetylated and coumaroylated glucosides were additionally identified. ANOVA of anthocyanin data revealed high malvidin content, as expected. Post hoc analyses showed that malvidin-3-glucoside (with the largest concentration amongst all the treatment groups) was significantly higher in wines from Site 2, Site 3 and Site 4. 

HPLC analysis for procyanidins were also considered leading to the identification of 15 polyphenolic compounds. No differences were found in the dimer digallate levels among the five wines (Table 5). ANOVA revealed the highest contents of gallic acid, procyanidin B1 and catechin polyphenolic compounds. While the highest levels were detected for procyanidin B1, the amounts were significantly different among all the wines examined. Wines from Site 1 contained the highest amount of gallic acid, followed by wines from Site 2–Site 4 and Site 3–Site 5. Site 1 wines are markedly different from those derived from Site 2–Site 4 and Site 3–Site 5. Differences between the latter two groups were also observed. The patterns for catechin were like those of procyanidin B1. The concentrations of procyanidin B2 and B4 were higher in wine samples from Site 4.

### 3.2. Estimation of Polyphenol Intake in Humans

Prior to determining the influence of polyphenol intake, the effects of alcohol-derived calories from all wines were examined (see Equations in Table 6). Atwater factor (7 kcal/g alcohol) was used for the calculation of alcohol-derived energy. Average height and weight values of individuals from the Spanish population were obtained from the Ministry of Health (Ministerio de Sanidad), estimated as 78 kg and 171 cm for men and 67 kg and 160 cm for women. Body mass index (BMI) was used to determine whether the Spanish population could be classified as normal weight, overweight, obese, or extremely obese. The values obtained indicated normal average weight of the population under study. A moderate activity level for both genders (shown in Table 6) was estimated, representing an activity factor of 1.78 for men and 1.64 for women, as recommended by the World Health Organization (WHO). Total calorie requirements were calculated from the Harris–Benedict equation [78]. Calorie intake due to metabolized alcohol and % energy due to alcohol were additionally calculated, along with blood alcohol levels/day. 

The recommended amounts for daily alcohol intake are 30–40 g for men and 10–20 g for women. Therefore, three levels of g alcohol/day were considered: 30, 35, and 40 for men (Table 7); and 10, 15, and 20 for women (Table 8). Considering the alcohol by volume percentage of the five different wines, the percentage of energy from alcohol consumption was calculated. Values ranging from 8.9% to 11.8% of total energy needs for men and 3.4% to 6.7% for women were obtained. The three dose ranges allowed for the estimation of average alcohol-derived energy. 

Levels of procyanidin and anthocyanidin absorption were calculated considering the recommended healthy threshold of wine intake per day. The % serum-available recovery of anthocyanin and procyanidin was estimated based on a recent review (3.9% for anthocyanins and 7.2% for procyanidins) [76]. The final serum concentration of polyphenols (either procyanidins or anthocyanidins) was evaluated considering the three levels of wine (determined earlier as 30, 35, and 40 g for men and 10, 15, and 20 g for women). A total daily dietary polyphenol intake of 200 mg/day was estimated, and the final estimated plasma concentration was calculated as mg/day. 

Data from Table 9a–d indicate that wines contribute to 2.9–17.1% of daily total procyanidin and 9.8–57.7% of daily total anthocyanin intake. Wines significantly contribute to total polyphenol intake (200 mg/day). Men ingested between 1.0 and 2.0 mg/day of procyanidins with all the wines, while women could only achieve this concentration in three of the wines (Site 1, Site 2, and Site 4). The concentration in men reached >2 mg/day for three of the wines (Site 1, Site 2, and Site 4), with higher procyanidin contents. On the other hand, if we consider the wine/dose, men only obtained >2 mg/day procyanidin with 35 and 40 g of Site 1, 40 g of Site 2, and 40 g of Site 4. In this case, men need to consume higher doses of wine to acquire >2 mg procyanidins/day. Women could not obtain >2 mg/day in any of the cases. In terms of anthocyanin composition in serum for women, the values depended on site wine and wine/dose, while men achieved >2 mg/day with all wines. Women achieved a concentration of <2 mg/day anthocyanin with Site 1 wine and >2 mg/day with the other wines (Site 2, Site 3, Site 4, and Site 5); however, it required an intake of higher volumes of wine.

Wines from Site 1 exhibited the highest tannin concentration (109.9 mg/L), the lowest anthocyanin amount (251.1 mg/L), and the highest alcohol content (16.1% alc. vol) among all the wines. The second highest level of procyanidin (86.2 mg/L) but lower alcohol content (14.5%) was found for Site 4 compared to the other wines. When considering the influence of mesoclimate and categorizing the wines into early region (E) and late region (L), no discrepancies in total amounts were discerned. In general, men exhibited over 2 mg/day of procyanidin and anthocyanin contents, while for both E and L groups, women showed a range of 1 to 2 mg/day.

## 4. Discussion

The bioavailability of wine-derived polyphenols in living organisms is notably limited; however, these polyphenols are closely linked to the beneficial effects of wine on human health. The potential health impact of varying polyphenol compositions in different types of wine, particularly in accordance with a recommended intake level that promotes health, remains to be fully understood. Establishing traceability of food, particularly in the context of esteemed quality wines related to specific vineyards, is a significant objective, as it facilitates the precise determination of food composition in daily dietary intake. Polyphenol intake from wine, like other dietary sources, is contingent upon the total content of these substances. Notably, the recommended amount of wine for healthy individuals is constrained due to the adverse effects of alcohol consumption. In this study, we conducted a comprehensive assessment of the potential health effects of diverse ranges of recommended wine intake, considering gender and age as factors. Both women and men were theoretically assessed based on divergent recommended dietary intake levels. Our primary aim was to determine the impact of varied wine compositions on the estimated polyphenol consumption in humans and total calorie intake. This investigation involved an extensive analysis of five wines to discern how distinct intake levels influence specific polyphenol patterns in the serum as well as calories due to alcohol. Additionally, the influence of altitude and sun exposure on the polyphenol composition of the five wines was considered. Furthermore, the caloric effect of alcohol was calculated and various wine doses were examined to elucidate the impact of the alcohol content of each wine on total dietary calories due to wine consumption.

Considering the low absorption of phenolic compounds, we postulated that certain wines might elicit equivalent polyphenol levels in the serum of healthy individuals when consumed in the recommended amounts daily. The selection of a specific wine may mitigate the effects of increased alcohol content. In this assessment, the caloric impact of alcohol ranged from 210 to 280 kcal for men and 70 to 140 kcal for women concerning total calorie requirements (2366 kcal for men and 2078 kcal for women). This is a salient consideration, as calories derived from alcohol account for 8.9% and 11.8% (minimum and maximum, respectively) of the total calorie requirement for men and 3.4% and 6.7% for women. It is to be highlighted that the wine choice is of notable importance, given that it was assessed that Site 1, Site 2, and Site 4 wines yielded elevated levels of procyanidins at a higher alcohol concentration (40 g) in both men and women, while Site 4 wine exhibited a lower alcohol concentration. As all wines possess varying alcoholic strengths, notably Site 1 and Site 4, the overall alcohol intake and total calorie count could be reduced without substantially impacting the acquisition of polyphenols from each wine. All wines/doses of wine with higher alcohol by volume (Site 2, Site 3, Site 4, and Site 5) presented increased concentrations of serum anthocyanin, with Site 4 wine being concurrently linked to a lower alcohol content. When choosing a wine, it is crucial to consider its components, particularly the polyphenols and alcohol content, as they have a significant impact on the quantity of phenols and the caloric load of the final product. The presence of polyphenols in wine is influenced by various factors, including the grape variety used, soil characteristics, climatic conditions, and winemaking methods. 

## 5. Conclusions

In summary, wines from different geographical viticultural areas tend to contain higher concentrations of polyphenols, which are associated with their reputation for high quality [79]. Additionally, the alcohol content can significantly differ among different wines, emphasizing the importance of carefully considering the nutritional and caloric profiles available on new wine labels to select products that align with individual preferences and dietary needs. It is suggested that depending on the healthy recommended servings of wine for either women or men, different wines can be selected to obtain similar amounts of procyanidins and anthocyanins in the diet. In addition, the selection of specific wines may avoid additional alcohol intake. The geographical wine production areas that lead to higher quality wines would be the preferred ones when considering a greater contribution of polyphenols. However, according to the new regulations, considering the number of calories from alcohol would be a determining factor for the consumer. In fact, the new future labels will allow us to understand the impact of wine quality on health without the wine necessarily being produced in a protected designation of origin; rather, the ingredients will determine an essential part of the wine’s quality. This finding is of great importance because alcohol-derived energy is not usually considered in total energy requirements, which is essential for patients with obesity or diabetes who need to control weight and energy expenditure due to carbohydrates, respectively. 

Our preliminary research serves as a foundation for future interventional studies aimed at accurately evaluating the effects of wine produced in specific grape-growing regions on human health. Within the context of the recent new labeling regulations in the wine sector, this study contributes additional value in tandem with the new EU labeling regulations, facilitating a deeper understanding of the importance of nutritional information in wine. Furthermore, while the findings support the beneficial effects of moderate alcohol consumption, it is imperative to acknowledge the significant toxicity of alcohol and the potential for dependency when consumed in large quantities, which applies to a non-negligible percentage of individuals. Consequently, we do not advocate alcohol consumption by non-drinkers due to the risk of triggering dependence or excessive intake. Moderate alcohol drinkers are advised to maintain this habit without surpassing the recommended healthy dose (up to 30–40 g/day in men and 10–20 g/day in women), provided there are no medical contraindications.

## Figures and Tables

**Figure 1 foods-13-00295-f001:**
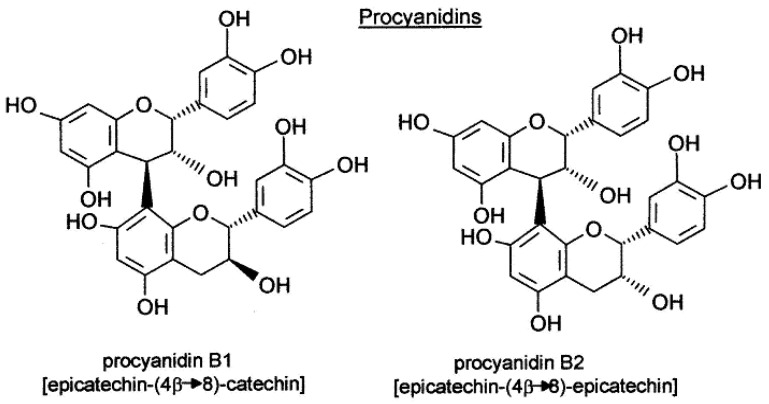
Procyanidin structure. The different subunits are linked by C4–C8 and, to a lesser extent, C4–C6 inter flavan bonds.

**Figure 2 foods-13-00295-f002:**
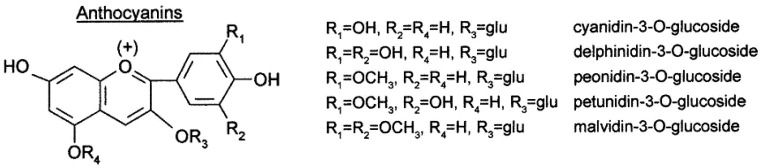
Anthocyanin structure.

**Figure 3 foods-13-00295-f003:**
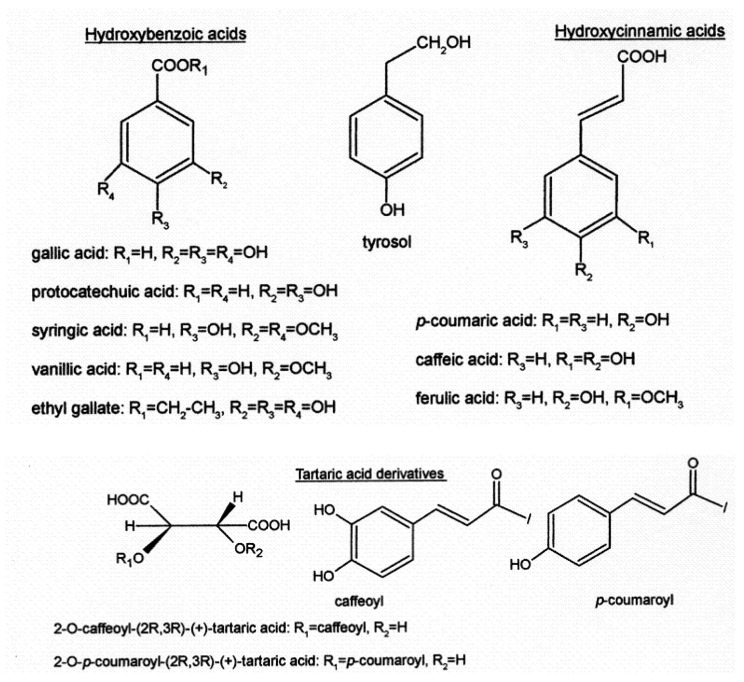
The main phenolic acids are also found in wine: hydroxybenzoic acid, tyrosol, hydroxycinnamic acid, tartaric acid, and derivatives.

**Figure 4 foods-13-00295-f004:**
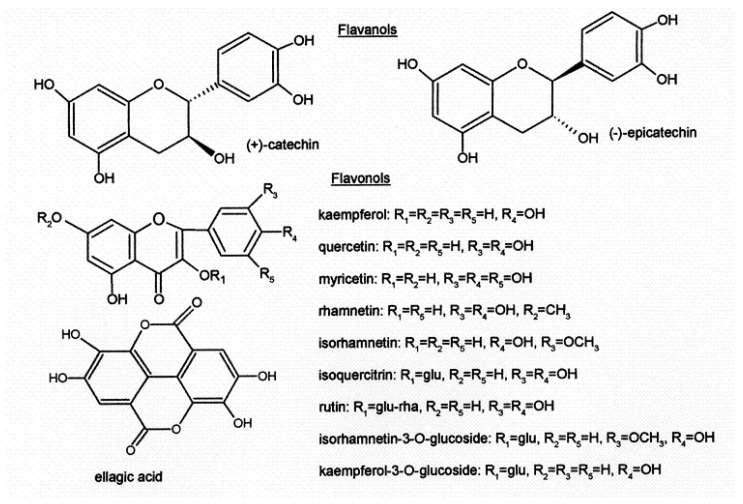
Structure of flavanols, flavonols and ellagic acid.

**Figure 5 foods-13-00295-f005:**
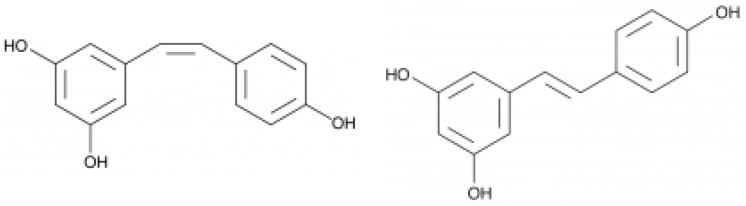
Structure of cis (**left**) and trans (**right**) resveratrol.

**Figure 6 foods-13-00295-f006:**
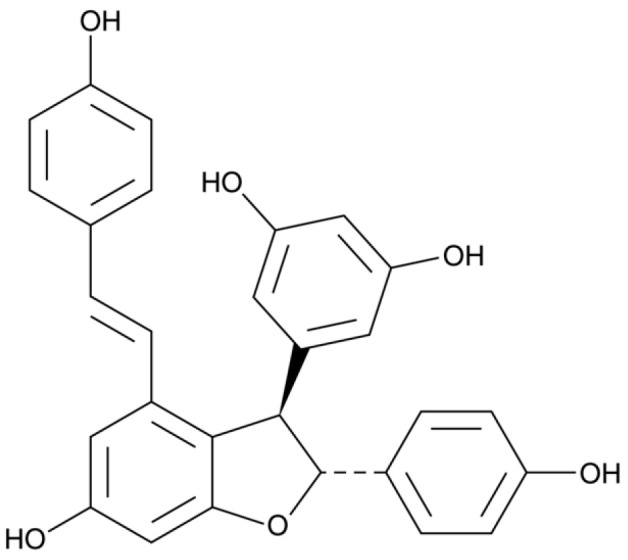
Structure of dimer trans-ε-viniferin (2) components of wine.

**Table 1 foods-13-00295-t001:** Generic classification of phenolic compounds.

Class of Wine Antioxidants	Compound
Flavanols	(+)-catechin, (-)-epicatechin
Hydroxybenzoic acids	gallic acid, protocatechuic acid, syringic acid vanillic acid, ethyl gallate, ellagic acid
Hydroxycinnamic acids	p-coumaric acid, o-coumaric acid, caffeic acid, ferulic acid
Tartaric acid and derivatives	caftaric acid (2-O-caffeoyl-(2R,3R)-(+)-tartaric acid) fertaric acid (2-O-feruloyl-(2R,3R)-(+)-tartaric acid) coutaric acid (2-O-p-coumaryl-(2R,3R)-(+)-tartaric acid)
Proanthocyanins	procyanidin B1, procyanidin B2
Phenols	tyrosol, hydroxytyrosol, 4-ethylguaiacol, tryphtophol
Flavonols	kaempherol, quercetin, rhamnetin, isorhamnetin, myricetin, kaempherol-3-O- glucoside, isorhamnetin-3-O-glucoside, isoquercitrin, rutin
Anthocyanins (coumaroylated, acylated, pyranoanthocyanins)	cyanidin-3-O-glucoside, delphinidin-3-O-glucoside, peonidin-3-O-glucoside, petunidin-3-O-glucoside, malvidin-3-O-glucoside, Vitisin A, Vitisin B
Resveratrols	*cis*-resveratrol, *trans*-resveratrol, *trans*-piceid, *cis*-piceid, *trans*-ε-viniferin α-viniferin

**Table 2 foods-13-00295-t002:** Phenolic compounds in different parts of grape and its products.

Origin	Phenolic Compounds
Seed	gallic acid, (+)-catechin, epicatechin, dimeric procyanidin, proanthocyanins
Skin	proanthocyanins, ellagic acid, myricetin, quercetin, kaempferol, trans-resveratrol
Leaf	myricetin, ellagic acid, kaempferol, quercetin, gallic acid
Stem	rutin, quercetin 3-*O*-glucuronide, trans-resveratrol, astilbin
Raisin	hydroxycinnamic acid, hydroxymethylfurfural
Red wine	malvidin-3-glucoside, peonidin-3-glucoside, cyanidin-3-glucoside, petunidin-3-glucoside, catechin, quercetin, resveratrol, hydroxycinnamic acid

**Table 3 foods-13-00295-t003:** Total anthocyanins (Total ANT), tannins, DMACH, pH, total titratable acidity (TTA), and alcohol by volume % (ABV) of each wine sample. Values with different letters denote a statistically (*p* < 0.05) significant difference. Mean and standard deviation.

Wine	Site 1	Site 2	Site 3	Site 4	Site 5
Total ANT (mg/L)	398.4 ± 9.6 a	466.6 ± 4.0 b	470.5 ± 22.0 b	432.9 ± 22.8 ab	451.5 ± 6.3 b
Tannins (g/L)	2.9 ± 0.2 c	2.4 ± 0.2 a	2.2 ± 0.3 a	1.7 ± 0.1 b	1.4 ± 0.1 b
DMACH (mg/L)	388.8 ± 22.9 a	361.4 ± 14.4 a	324.0 ± 7.9 b	295.6 ± 8.0 bc	263.1 ± 4.5 c
pH	3.17 ± 0.1 ac	3.27 ± 0.0 abc	3.28 ± 0.0 abc	3.35 ± 0.0 b	3.21 ± 0.0 c
TTA (g/L)	6.63 ± 0.2 ac	6.40 ± 0.2 c	6.39 ± 0.0 c	6.85 ± 0.1 a	7.62 ± 0.1 b
ABV (%)	16.10 ± 0.4 a	15.20 ± 0.6 abc	14.80 ± 0.1 bde	14.50 ± 0.2 ce	15.40 ± 0.2 ad

**Table 4 foods-13-00295-t004:** HPLC analysis of anthocyanin composition (ppm). Values with different letters denote a statistically (*p* < 0.05) significant difference. Mean and standard deviation.

Wine	Site 1	Site 2	Site 3	Site 4	Site 5
malvidin-3-G	164.1 ± 4.8 a	217.1 ± 4.5 d	222.6 ± 6.3 cd	203.6 ± 7.4 b	226.8 ± 1.8 c
petunidin-3-G	7.6 ± 0.4 a	8.8 ± 2.3 b	7.7 ± 2.0 a	5.5 ± 1.8 a	6.9 ± 2.0 a
delphinidin-3-G	2.0 ± 1.7 a	2.7 ± 1.0 b	2.2 ± 1.8 a	2.0 ± 1.3 a	2.6 ± 0.6 a
peonidin-3-G	4.6 ± 0.2 b	5.2 ± 0.1 d	5.5 ± 0.1 e	3.5 ± 0.0 a	4.7 ± 0.0 c
cyanidin-3-G	0.3 ± 0.0 b	0.3 ± 0.1 ab	0.3 ± 0.0 b	0.2 ± 0.0 a	0.3 ± 0.0 b
malvidin-3-AG	32.3 ± 1.6 a	57.1 ± 1.9 c	54.8 ± 0.6 b	59.5 ± 1.8 c	64.6 ± 1.5 d
petunidin-3-AG	0.8 ± 0.0 a	1.1 ± 0.0 d	1.0 ± 0.0 c	0.9 ± 0.0 b	0.9 ± 0.0 b
delphinidin-3-AG	0.3 ± 0.0 a	0.4 ± 0.0 b	0.3 ± 0.0 a	0.3 ± 0.0 a	0.3 ± 0.0 a
peonidin-3-AG	1.5 ± 0.0 a	1.8 ± 0.0 b	1.9 ± 0.0 c	1.5 ± 0.0 a	2.0 ± 0.1 d
cyanidin-3-AG	0.2 ± 0.0	0.2 ± 0.0	0.2 ± 0.0	0.2 ± 0.0	0.2 ± 0.0
malvidin-3-CG	28.7 ± 2.4 a	38.5 ± 5.6 b	41.5 ± 1.2 b	34.6 ± 7.9 ab	37.3 ± 7.6 ab
petunidin-3-CG	3.9 ± 0.2 a	5.8 ± 0.0 c	4.4 ± 0.2 b	3.7 ± 0.2 a	4.2 ± 0.1 b
delphinidin-3-CG	1.3 ± 0.1 b	1.8 ± 0.0 c	1.2 ± 0.0 b	1.0 ± 0.1 a	1.0 ± 0.0 a
peonidin-3-CG	2.5 ± 0.3	3.0 ± 0.4	2.8 ± 0.1	2.0 ± 0.2	2.8 ± 0.0
cyanidin-3-CG	1.1 ± 0.0 c	1.2 ± 0.0 d	1.0 ± 0.0 c	0.7 ± 0.0 a	0.9 ± 0.0 b
Total Anthocyanins (mg/L)	251.1 ± 11.7 a	345.0 ± 15.9 c	347.6 ± 12.3 c	319.1 ± 20.7 b	355.5 ± 13.7 c

**Table 5 foods-13-00295-t005:** HPLC analysis of five red wines leading to the identification of 15 polyphenolic compounds (ppm). Values with different letters denote a statistically (*p* < 0.05) significant difference. Mean and standard deviation.

Wine	Site 1	Site 2	Site 3	Site 4	Site 5
procyanidin B3	3.5 ± 0.2 c	2.4 ± 0.1 b	2.3 ± 0.2 b	2.7 ± 0.1 b	2.0 ± 0.1 a
procyanidin B1	23.7 ± 0.0 e	19 ± 0.2 d	13.7 ± 0.3 c	14.5 ± 0.1 b	12.6 ± 0.2 a
procyanidin B4	8.5 ± 0.2 d	6.3 ± 0.0 c	6.9 ± 0.1 b	9.5 ± 0.2 a	7.1 ± 0.2 b
procyanidin B2	8.6 ± 0.1 d	6.3 ± 0.0 c	6.9 ± 0.1 b	9.6 ± 0.1 a	7.1 ± 0.1 b
dimer monogallate	1.7 ± 0.1 c	2.5 ± 0.1 a	3.0 ± 0.1 b	3.1 ± 0.2 b	1.9 ± 0.0 c
procyanidin B 5.1	1.2 ± 0.1 b	1.2 ± 0.1 b	0.9 ± 0.0 a	0.8 ± 0.1 a	0.8 ± 0.0 a
dimer digallate	0.0 ± 0.0	0.0 ± 0.0	0.0 ± 0.0	0.0 ± 0.0	0.0 ± 0.0
procyanidin B 6.6	3.0 ± 0.2 c	3.6 ± 0.1 b	4.2 ± 0.2 a	4.7 ± 0.1 a	4.2 ± 0.0 a
gallic acid	22.1 ± 0.3 d	16 ± 0.6 c	14.6 ± 0.5 a	15.3 ± 0.5 cb	13.8 ± 0.2 a
catechin	10.7 ± 0.2 e	7.7 ± 0.1 d	5.0 ± 0.1 c	5.4 ± 0.1 b	4.7 ± 0.1 a
epicatechin	3.2 ± 0.1 b	1.8 ± 0.1 a	2.2 ± 0.0 a	3.1 ± 0.0 b	2.6 ± 0.0 a
epicatechin gallate	0.1 ± 0.1 b	0.1 ± 0.0 b	0.1 ± 0.0 b	0.1 ± 0.0 b	0.0 ± 0.0 a
trimer C 0.6	7.4 ± 0.3 b	6.6 ± 0.3 b	5.7 ± 0.1 a	5.5 ± 0.5 a	5.0 ± 0.2 a
trimer C 2.4	9.8 ± 0.1 e	6.9 ± 0.0 d	4.8 ± 0.2 c	5.5 ± 0.1 b	4.2 ± 0.1 a
trimer C1	6.4 ± 0.2 b	4.1 ± 0.2 a	4.5 ± 0.2 a	6.6 ± 0.2 b	4.4 ± 0.3 a
TOTAL	109.9 ± 1.0 c	84 ± 0.7 b	74.9 ± 1.0 a	86.2 ± 0.9 b	70.3 ± 0.5 a

**Table 6 foods-13-00295-t006:** Values and equations used for calculation of daily energy needs, alcohol energy, % energy due to alcohol consumption, healthy wine (recommended volume of wine intake) and blood alcohol level (BAL). The % energy due to alcohol consumption is based on daily energy needs determined using Harris–Benedict Equation.

Table Calculation Equations		
Description	Unit	Calculation
Activity factor (physical activity level, PAL) (World Health Organization)	No	Light level activity factor: 1.55 (men) and 1.56 (women) Moderate level activity factor: 1.78 (men) and 1.64 (women) Intense level activity factor: 2.10 (men) and 1.82 (women)
Daily Energy Needs, REE (Harris–Benedict Equation)	kcal	Women: REE = 655.1 + 9.56 × weight (kg) + 1.85 × height (cm) − 4.68 × age (years) Men: REE = 66.5 + 13.75 × weight (kg) + 5.0 × height (cm) − 6.78 × age (years) +20% light; +30% moderate and 50–75% intense activity +10% Food-Induced Thermogenesis
Alcohol Energy	kcal	mL wine/day·(%alc. vol/100) × alcohol density × 7 Kcal/g Ethanol 7 kcal/g alcohol (Atwater Factor)
% Energy due to alcohol consumption	%	% = Alcohol energy/Daily Energy Needs
Healthy wine (intake of wine volume recommended)	mL/day	V (mL wine) = (g healthy alcohol/ % DA × 0.789)
Blood alcohol level (BAL)	g/day	BAL = (g healthy alcohol/kg) × (60 L/100 kg)

**Table 7 foods-13-00295-t007:** Estimated calculation for men based on an average weight (78.1 kg), average high (171 cm), average age (45 years old), body mass index (22.8), activity level (moderate), activity factor 1.78 and daily energy needs determined using Harris–Benedict Equation (2366 Kcal).

	Men														
Wine	Site 1			Site 2			Site 3			Site 4			Site 5		
Healthy alcohol serving (g/day)	30	35	40	30	35	40	30	35	40	30	35	40	30	35	40
Alcohol Energy (Kcal)	210	245	280	210	245	280	210	245	280	210	245	280	210	245	280
Healthy wine (mL/day)	233	272	311	247	288	329	253	296	338	259	302	345	244	284	325
% alc. Vol	16.1			15.2			14.8			14.5			15.4		
Blood alcohol levels/day	0.23	0.27	0.31	0.23	0.27	0.31	0.23	0.27	0.31	0.23	0.27	0.31	0.23	0.27	0.31
% Energy due to alcohol consumption	8.9	10.4	11.8	8.9	10.4	11.8	8.9	10.4	11.8	8.9	10.4	11.8	8.9	10.4	11.8

**Table 8 foods-13-00295-t008:** Estimated calculation for women based on an average weight (67.1 kg), average high (160 cm), average age (45 years old), body mass index (21), activity level (moderate), activity factor 1.64 and daily energy needs determined using Harris–Benedict Equation (2078 Kcal).

	Women														
Wine	Site 1			Site 2			Site 3			Site 4			Site 5		
Healthy alcohol serving (g/day)	10	15	20	10	15	20	10	15	20	10	15	20	10	15	20
Alcohol Energy (Kcal)	70	105	140	70	105	140	70	105	140	70	105	140	70	105	140
Healthy wine (mL/day)	78	117	156	82	123	164	84	127	169	86	129	172	81	122	162
% alc. Vol	16.1			15.2			14.8			14.5			15.4		
Blood alcohol levels/day	0.09	0.13	0.18	0.09	0.13	0.18	0.09	0.13	0.18	0.09	0.13	0.18	0.09	0.13	0.18
% Energy due to alcohol consumption	3.4	5.1	6.7	3.4	5.1	6.7	3.4	5.1	6.7	3.4	5.1	6.7	3.4	5.1	6.7

**Table 9 foods-13-00295-t009:** (**a**) Estimated healthy wine dose and % gastrointestinal absorption for men: 3.9% for anthocyanins (ANT). Results are presented as mg/day in three groups, specifically, <1 mg/day, 1 < x < 2 mg/day and >2 mg/day. (**b**) Estimated healthy wine dose and % gastrointestinal absorption for men: 7.2% for phenols (PRO). Results are presented as mg/day in three groups, specifically, <1 mg/day, 1 < x < 2 mg/day and >2 mg/day. (**c**) Estimated healthy wine dose and % gastrointestinal absorption for women: 3.9% for anthocyanins (ANT). Results are presented as mg/day in three groups, specifically, <1 mg/day, 1 < x < 2 mg/day and >2 mg/day. (**d**) Estimated healthy wine dose and % gastrointestinal absorption for women: 7.2% for phenols (PRO). Results are presented as mg/day in three groups, specifically, <1 mg/day, 1 < x < 2 mg/day and >2 mg/day.

(**a**)									
		**Total ANT. (mg/L)**	**ANT.** **Intake (mg/day)**	**% BIO. ANT.**	**ANT. Intake** **(%)**	**ANT.** **Serum** **Intake** **(mg/day)**	**Serum Intake**		
							**<1**	**1 < x < 2**	**>2**
Men	Site 1	251.1	58.6	3.9	29.3	2.29			x
			68.3		34.1	2.66			x
			78.1		39.0	3.05			x
	Site 2	345	85.2		42.6	3.32			x
			99.4		49.7	3.88			x
			113.5		56.8	4.43			x
	E (early ripeness)	298.1	71.9		36.0	2.81			x
			83.8		41.9	3.27			x
			95.8		47.9	3.74			x
	Site 3	347.6	87.9		44.0	3.43			x
			102.9		51.4	4.01			x
			117.5		58.7	4.58			x
	Site 4	319.1	82.6		41.3	3.22			x
			96.4		48.2	3.76			x
			110.1		55.0	4.29			x
	Site 5	355.5	86.7		43.4	3.38			x
			101.0		50.5	3.94			x
			115.5		57.8	4.51			x
	L (late ripeness)	340.7	85.8		42.9	3.35			x
			100.1		50.0	3.90			x
			114.4		57.2	4.46			x
(**b**)									
		**TOT. PRO (mg/L)**	**PRO.** **Intake** **(mg/day)**	**%** **BIO. PRO.**	**PRO.** **Intake** **(%)**	**PRO.** **Serum Intake** **(mg/day)**	**Serum Intake**		
							**<1**	**1 < x < 2**	**>2**
Men	Site 1	109.9	25.7	7.2	12.8	1.85		x	
			29.9		14.9	2.15			x
			34.2		17.1	2.46			x
	Site 2	84.4	20.8		10.4	1.50		x	
			24.3		12.2	1.75		x	
			27.8		13.9	2.00			x
	E (early ripeness)	97.2	23.3		11.6	1.67		x	
			27.1		13.6	1.95		x	
			31.0		15.5	2.23			x
	Site 3	74.9	18.9		9.5	1.36		x	
			22.2		11.1	1.60		x	
			25.3		12.7	1.82		x	
	Site 4	86.2	22.3		11.2	1.61		x	
			26.0		13.0	1.87		x	
			29.7		14.9	2.14			x
	Site 5	70.3	17.2		8.6	1.24		x	
			20.0		10.0	1.44		x	
			22.8		11.4	1.65		x	
	L (late ripeness)	77.1	19.5		9.7	1.40		x	
			22.7		11.4	1.64		x	
			26.0		13.0	1.87		x	
(**c**)									
		**TOT. ANT. (mg/L)**	**ANT.** **Intake** **(mg/day)**	**% BIO. ANT**	**ANT.** **Intake** **(%)**	**ANT.** **Serum Intake (mg/day)**	**Serum Intake**		
							**<1**	**1 < x < 2**	**>2**
Women	Site 1	251.1	19.6	3.9	9.8	0.76	x		
			29.4		14.7	1.15		x	
			39.2		19.6	1.53		x	
	Site 2	345	28.3		14.1	1.10		x	
			42.4		21.2	1.65		x	
			56.6		28.3	2.21			x
	E (early ripeness)	298.1	23.9		12.0	0.93	x		
			35.9		18.0	1.40		x	
			47.9		23.9	1.87		x	
	Site 3	347.6	29.2		14.6	1.14		x	
			44.1		22.1	1.72		x	
			58.7		29.4	2.29			x
	Site 4	319.1	27.4		13.7	1.07		x	
			41.2		20.6	1.61		x	
			54.9		27.4	2.14			x
	Site 5	355.5	28.8		14.4	1.12		x	
			43.4		21.7	1.69		x	
			57.6		28.8	2.25			x
	L (late ripeness)	340.7	28.5		14.2	1.11		x	
			42.9		21.4	1.67		x	
			57.1		28.5	2.23			x
(**d**)									
		**TOT. PRO. (mg/L)**	**PRO.** **Intake** **(mg/day)**	**% BIO. PRO.**	**PRO** **Intake** **(%)**	**PRO** **Serum Intake (mg/day)**	**Serum PRO. EXP. Intake**		
							**<1**	**1 < x < 2**	**>2**
Women	Site 1	109.9	8.6	7.2	4.3	0.62	x		
			12.9		6.4	0.93	x		
			17.1		8.6	1.23		x	
	Site 2	84.4	6.9		3.5	0.50	x		
			10.4		5.2	0.75	x		
			13.8		6.9	1.00		x	
	E (early ripeness)	97.2	7.7		3.9	0.56	x		
			11.6		5.8	0.84	x		
			15.5		7.7	1.12		x	
	Site 3	74.9	6.3		3.1	0.45	x		
			9.5		4.8	0.68	x		
			12.7		6.3	0.91	x		
	Site 4	86.2	7.4		3.7	0.53	x		
			11.1		5.6	0.80	x		
			14.8		7.4	1.07		x	
	Site 5	70.3	5.7		2.8	0.41	x		
			8.6		4.3	0.62	x		
			11.4		5.7	0.82	x		
	L (late ripeness)	77.1	6.5		3.2	0.47	x		
			9.7		4.9	0.70	x		
			13.0		6.5	0.93	x		

## Data Availability

Not applicable.

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
