# Peer review of "New Labeling Rules for Wine: Wine Alcohol-Derived Calories and Polyphenol Consumption on Health"

_foods, 2024, doi:10.3390/foods13020295_

Round 1
Reviewer 1 Report
Comments and Suggestions for Authors
In the review submitted for publication in the journal Foods, the authors discuss the latest regulations for the correct labeling of wine beverages, both in terms of calorie content and biologically active substances. They describe the contributions of different categories of phenolic substances to the composition of wine, depending on the type and geographical origin of the grapes used in its production. The results of the discussion are clear and comprehensive, including the conclusions. However, the manuscript could be improved to make it easier to read.
Comments:
- The literature does not seem to follow the instructions of the journal Foods.
- The English language needs to be revised due to some typing errors.
- Tables 1-2. The authors could add the references in the tables under the substances or their origin.
- Tables 3-5. References could also be added to this table.
- ANOVA is certainly a good method for analyzing variance and significance of data. However, the authors might consider using a multivariate data analysis procedure, such as principal component analysis (PCA), to describe better the data pattern and the results of their investigation. There are different publications regarding these topics that could be considered by the Authors in the revision step, for example:
https://doi.org/10.1016/j.lwt.2022.114004
https://doi.org/10.3390/antiox12030622
Author Response
Dear Reviewer,
I would like to thank you for his/her suggestions which I believe greatly improved this manuscript. The issues pointed out by the reviewer have been addressed as listed in what follows.
-The literature has been updated to the Food journal format as required.
-English spelling and grammar has been amended and improved
-The introduction has been fully improved, particularly giving more details on the new labelling. Also, the discussion and conclusions have been improved, connecting the findings with the awareness of the wine consumers.
I considered doing a PCA but unfortunaltely I have not enough data, I should have some more wines to better describe the pattern, in this case the objective was to highlight the importance of 5 wines, as this research is related to nutrition facts of labels due to the upcoming wine label regulation. However, I mentioned that this is the basis for future studies related to this subject.
References on the tables have been added. I also cited one of your references in the text (line 790).
I very much appreciated your suggestions.
Yours faithfully,
Reviewer 2 Report
Comments and Suggestions for Authors
In the manuscript by Antoni et al. entitled »New labelling rules for wine: wine alcohol-derived calories and polyphenol consumption on health«, the authors investigate the effects of red wine consumption on health when the effects of alcohol consumption on energy intake and polyphenol consumption on health are considered. The paper is well written, and in general the conclusions (taking into account alcohol and polyphenol intake) seem correct.
The minor problem I have is the concept of the study. Namely, I cannot see a direct link between the title of the paper (which starts with »New labelling rules for wine: ...«) and the new labelling rules for the wine sector and aromatized wine products mentioned in the last lines of the Abstract and Conclusions. So, there is no information on what exactly is included in these new labelling rules. I ask the authors to improve the paper by better linking the title (and content) of the paper to the new European labelling rules. Therefore, please also add the reference to the new European labelling rules and describe in more detail how the study relates to these rules.
Several technical improvements are also necessary:
1.) The part where the references are described needs to be thoroughly revised and unification of the way the references are given. For example, Ref. 1 is given in a way that is completely different from the way other references are described. Ref. 44 contains part of the data that goes to Ref. 68, different denominators are used (sometimes semicolons and sometimes commas), sometimes full names are given for journals and sometimes abbreviations, ... Please standardise this part.
2.) The remnants of the template for the reference section (Refs. 80-87) should be removed.
3.) In the Author Contributions part there is a remnant from the template (»X.X.« - line 693) which should be removed.
4.) The decimal separator used in English is a period and not a comma. This is particularly annoying when the same separators occur in one and the same equation (see, for example, the equations in Table 6). There are also many other places where the decimal separator should be corrected, so please check the entire manuscript.
5.) In Table 6, when stating the Harris-Benedict equation, the units in which weight, height (not high – please change) and age should be inserted should also be stated.
6.) There is no reference to the quantitative significance of the activity factors (Table 6) (light, moderate and intense level are only descriptions).
7.) Also, when units are given next to the numbers, a space is used between the number and the symbol to which it refers. This rule is sometimes followed (e.g. line 579) and sometimes not (line 582). Please check the entire manuscript.
Comments on the Quality of English LanguageMinor editing of English language required
Author Response
Dear Reviewer,
I would like to thank you for his/her suggestions which I believe greatly improved this manuscript. The issues pointed out by the reviewer have been addressed as listed in what follows.
-The literature has been updated to the Food journal format required.
-English spelling and grammar have been amended an improved.
-The remnants of the template have been removed.
-The decimal separators have also been amended.
-The introduction has been modified, particularly to introduce the subject on the new labelling, and also during the discussion and conclusions has been improved the context related to the importance of the new labelling for the awareness of the wine consumers.
-Table 6 has been added the units and added the reference.
I very much appreciated your suggestions.
Yours faithfully,
Reviewer 3 Report
Comments and Suggestions for Authors
The publication is interesting and can be published after appropriate editing. Here are some notes.
1. The introduction should be reformatted as it is currently heavily defragmented. I believe it should start with some general information about obtaining wine, then smoothly move into the following sections.
2. The graphic representation of the various components of the wine (structural formulas) are redundant and burden the publication.
3. The type of publication must be specified - scientific or review. It does not sound logical to have materials and methods, results and discussion, etc. in a review publication.
Minor editing of English language required
Author Response
Dear Reviewer,
I would like to thank you for his/her suggestions which I believe greatly improved this manuscript. The issues pointed out by the reviewer have been addressed as listed in what follows.
-The introduction has been modified, particularly to introduce the subject on the new labelling, and during the discussion and conclusions has been improved the context related to the importance of the new labelling for the awareness of the wine consumers.
-Some of the figures describing the type of phenols in wine has been updated and some of them have been eliminated to avoid redundancy.
-English spelling and grammar has been amended an improved
-The paper is an article that shows the importance of the nutrition facts of the new labelling in wine, thus connecting the importance of a wine choice (from a specific area) to their impact on the phenols and calory intake.
I very much appreciated your suggestions.
Yours faithfully,
Round 2
Reviewer 1 Report
Comments and Suggestions for Authors
The review has been improved. Authors answered the reviewer's comments clearly and comprehensively.
Author Response
Dear Reviewer, I uploaded the final version, upon reviewing and updating references and some punctuation recommended by editor.
Much appreciate your suggestions as I think the review has improved substantially.
Please find attached last version.
Yours faithfully,